# Influence of the Medium Composition and the Culture Conditions on Surfactin Biosynthesis by a Native *Bacillus subtilis natto* BS19 Strain

**DOI:** 10.3390/molecules26102985

**Published:** 2021-05-18

**Authors:** Beata Koim-Puchowska, Grzegorz Kłosowski, Joanna Maria Dróżdż-Afelt, Dawid Mikulski, Alicja Zielińska

**Affiliations:** Department of Biotechnology, Kazimierz Wielki University, K.J. Poniatowski St12, 85-671 Bydgoszcz, Poland; koimpuch@ukw.edu.pl (B.K.-P.); jdrozdz@ukw.edu.pl (J.M.D.-A.); dawidmikulski@ukw.edu.pl (D.M.); alicja.zielinska@student.ukw.edu.pl (A.Z.)

**Keywords:** surfactin, *Bacillus subtilis*, medium components

## Abstract

An effective microbial synthesis of surfactin depends on the composition of the culture medium, the culture conditions and the genetic potential of the producer strain. The aim of this study was to evaluate the suitability of various medium components for the surfactin producing strain and to determine the impact of the culture conditions on the biosynthesis of surfactin isoforms by the newly isolated native strain *Bacillus subtilis natto* BS19. The efficiency of surfactin biosynthesis was determined by measuring the surface tension of the medium before and after submerged culture (SmF) and by qualitative and quantitative analysis of the obtained compound by high performance liquid chromatography. The highest efficiency of surfactin biosynthesis was achieved using starch as the carbon source and yeast extract as the nitrogen source at pH 7.0 and 37 °C. Potato peelings were selected as an effective waste substrate. It was shown that the increase in the percentage of peel extract in the culture medium enhanced the biosynthesis of surfactin (mg/L) (2–30.9%; 4–46.0% and 6–58.2%), while reducing surface tension of the medium by about 50%. The obtained results constitute a promising basis for further research on biosynthesis of surfactin using potato peelings as a cheap alternative to synthetic medium components.

## 1. Introduction

Surfactin analogs are one of the most effective surfactants. When used at a very low concentration of 20 μM, they reduce the surface tension of water from 72 to 22 mN/m. Moreover, this compound lowers the interfacial tension in the n-hexane/water system from 43 to <1 mN/m [1,2]; hence the wide application of this lipopeptide in many areas, including medicine, environmental protection and agriculture as an alternative to synthetic equivalents [3,4,5,6,7]. The properties of surfactin result from this amphiphilic structure. The surfactin molecule consists of a hydrophilic part, namely, a peptide ring consisting of 7 amino acids (l-glutamine, l-leucine, d-leucine, l-valine, l-asparagine, d-leucine and l-leucine) and a hydrophobic part, namely, β-hydroxy acid [8]. Microorganisms (mainly *Bacillus subtilis* strains) produce many isoforms (analogs) of this compound. The isoforms differ in the configuration of amino acids in the peptide ring, and in length (from 10 (C-10) to 16 (C-16)) and the branching of fatty acid side chains [9,10]. Importantly, the differences in the structure of surfactin are reflected in surfactant and foaming properties of this compound [11,12]. The biosynthesis of individual surfactin isoforms and thus the properties of the mixture are determined by the fermentation environment and the type and availability of nutrients [10,13,14]. Therefore, although the history of surfactin dates back to 1968, numerous studies are still being carried out on the influence of the composition of the culture medium and the culture conditions on the production of this compound [13,15,16,17,18,19]. The temperature of *B. subtilis* cultures producing surfactin ranges over quite a wide interval: from 25 to 37 °C [20]. The microbiological synthesis of this compound was also confirmed in the cultures incubated at temperatures higher than 40 °C [21]. The culture medium pH significantly changes the metabolism of microorganisms (including the transport of compounds through the cell membrane), and affects the intensity of surfactin production. It was shown that the neutral pH of the medium promoted the synthesis of surfactin [22,23]. *Bacillus* spp. strains prefer water-soluble nutrients; on the other hand, hydrophobic hydrocarbons can even lead to inhibition of surfactin synthesis [13,24,25]. Additionally, this choice is supported by the lower cost of the water-soluble medium components [5,24,26]. The source of carbon in synthetic media is mainly glucose, but also fructose, sucrose, maltose or soluble starch, which are preferred to glycerol or hexadecane [16,21,25,27]. Cooper’s mineral medium, along with Landy’s, is one of the most widely used culture mediums for surfactin biosynthesis and the basis for modifying the nutrient ratio [28,29]. This medium contains glucose (40 g/L) as a carbon source [13]. As it turns out, the choice of the carbon source is not the only significant influence on the biosynthesis of surfactin, the concentration of the substrate used is also important. A glucose concentration that is too high (60–80 mg/L) lowers the pH of the medium and thus inhibits the production of this biosurfactant [30]. At elevated glucose concentrations, bacterial cells probably do not show physiological adaptation to the biosynthesis of surfactants [27]. For the production of surfactin by *Bacillus* spp., the aforementioned Cooper’s medium and Landy’s medium are used, with two different nitrogen sources: ammonium nitrate and glutamic acid, respectively [29]. Ammonium chloride, ammonium nitrate, ammonium sulfate, ammonium ferric citrate, potassium nitrate, sodium nitrate, yeast and beef extract, peptone and tryptone are also used as nitrogen sources [21,31,32]. The study by Das and Mukherjee [21] on *B. subtilis* DM-03 and DM-04 strains demonstrated the importance of the proper selection of nitrogen source for surfactin biosynthesis. They showed that in media poor in nitrogen, surfactin with reduced surfactant properties may be produced. Moreover, the two tested strains differed in their nitrogen source preferences. Ammonium nitrate turned out to be better for the DM-03 strain, while tryptone was better for DM-04. On the other hand, Ghribi and Ellouze-Chaabouni [27] showed that among the organic nitrogen sources (urea, pancreatic casein extract, beef extract, yeast extract) used for the synthesis of biosurfactants by the *Bacillus subtilis SPT1* strain, urea was the most effective (750 mg/L of biosurfactants using 5 g/L of urea).

The high cost of the medium components prompted researchers to analyze waste products as substrates for *Bacillus* spp. cultures. The use of waste, in particular waste from agri-food processing, as substrates for the production of biologically active compounds is a rational way of managing substances that may pose a threat to the environment. At the same time, this solution allows reduction of the production costs of biodegradable surfactants, and thus makes them more affordable. Assessing the suitability of a substrate is an important part of the feasibility study of a biotechnological process. The components contained in the waste must guarantee the proper development of microorganisms and efficient synthesis of biosurfactants, and at the same time, the disposal of residues should not pollute the environment. Therefore, waste with a high carbohydrate or fat content is still being sought (Makkar and Cameotra [33]) that can replace synthetic sources of carbon and nitrogen, and at the same time contain essential minerals. So far, the following waste products have been used to produce biosurfactants synthesized by various strains of *Bacillus* spp.: molasses [32,34]; potato, orange and banana peel extract; whey [3 2]; soy flour and rice straw [35]; cassava wastewater [24]; and cashew juice [36]. It should be emphasized that the efficiency of waste substrates is determined by the metabolic abilities of individual strains and the conditions of the surfactin biosynthesis process.

The aim of the study was to analyze the effect of modifications of the Cooper’s medium composition (carbon source, nitrogen source) on the biosynthesis of surfactin (amount and percentage of individual isoforms) produced by the native strain of *B. subtilis natto* BS19. We assessed the growth rate of cell biomass in modified media and the reduction of the surface tension of the medium at various temperature and pH conditions. After verification and analysis of the usefulness of synthetic carbon sources in biosynthesis of various surfactin isoforms, we additionally investigated the possibility of using potato peel extract as an effective and cheaper carbon source for the producer strain.

## 2. Results

### 2.1. Synthetic Carbon Sources

The data presented in Table 1 show that not all of the analyzed carbon sources were assimilated by the native *Bacillus subtilis natto* BS19 strain. Of the sugars tested, lactose and xylose were not the preferred substrates. A slight increase in biomass was observed in the media containing these sugars (lactose: 0.1 ± 0.004 mg/L; xylose: 0.05 ± 0.009 mg/L). Surfactin was not produced. Therefore, no reduction in surface tension was observed (xylose) or the surface tension decreased only slightly (lactose). Among the remaining carbon sources, the most intensive surfactin biosynthesis was obtained with the use of soluble starch (133.38 ± 4.827 mg/L), although the increase in cell biomass (1.03 ± 0.173) in this medium was similar to that for other sugars (except for xylose and lactose). Moreover, for starch substrates, the reduction in surface tension was over 70%.

The hydrolytic activity tests carried out on the agar medium with 1% soluble starch addition (Figure 1) clearly confirmed that the strain assimilated this substrate better than sodium carboxymethyl cellulose (CMCNa) or tributyrin.

Glucose (monosaccharide) was the second-best source of carbon after starch. It also ensured the efficient production of surfactin analogues by *B. subtilis natto* BS19. The reduction in surface tension in the post-culture medium containing glucose, was approximately 45%. However, the concentration of surfactin obtained with glucose was less than 10 mg/L. The use of various carbon sources resulted in changes in the concentrations and relative amounts of individual isoforms synthesized by the studied strain (Figure 2).

Two surfactin isoforms were synthesized with arabinose and fructose; three with glycerin, maltose and mannose; five with galactose and cellobiose; and six with the other sugars (glucose, sucrose, sorbitol, soluble starch and trehalose). Among the produced surfactin isoforms, the analogs marked as C and E dominated. The concentration (mg/L) of the isoforms produced by the tested strain differed significantly depending on the carbon source used in the culture medium (surfactin A: from 0.03 (mannose) to 2.14 (soluble starch); surfactin B: from 0.13 (maltose) to 5.26 (soluble starch); surfactin C: from 0.11 (fructose) to 67.64 (soluble starch); surfactin D from 0.06 (sucrose) to 15.05 (soluble starch); surfactin E from 0.10 (arabinose) to 26.23 (soluble starch); surfactin F from 0.52 (glucose) to 17.04 (soluble starch)).

### 2.2. Synthetic Nitrogen Sources

Four organic substrates (beef extract, malt extract, peptone, yeast extract) and a mineral one (ammonium nitrate) were tested as nitrogen sources in the microbial biosynthesis of surfactin. The culture medium pH was 7.0 and the incubation temperature was 37 °C. The beef extract contributed little to the production of surfactin (Figure 3a–c).

Other sources, especially peptone and yeast extract, significantly promoted surfactin biosynthesis (peptone: 288.02 ± 1.04 mg/L, yeast extract: 368.36 ± 94.23 mg/L, as compared to ammonium nitrate: 133.38 ± 4.83 mg/L, malt extract: 50.10 ± 6.18 mg/L and beef extract: 7.42 ± 0.79 mg/L). For all nitrogen sources, except for the beef extract, a significant reduction (>50%) of the surface tension of the medium was observed. Supplementing the medium with peptone and yeast extract resulted in an average 4.5-times higher increase in bacterial biomass than with the use of ammonium nitrate or malt extract. On the other hand, when using the beef extract, the biomass concentration was more than 11 times lower than that in the medium with peptone or yeast extract. The use of yeast extract, malt extract or peptone promoted the biosynthesis of the E isoform. For ammonium nitrate and beef extract, the production of analogs C and F, respectively, was observed. For all nitrogen sources, we obtained either 5 (for yeast extract, peptone, beef extract) or 6 surfactin isoforms (for malt extract, ammonium nitrate) (Figure 4). The concentration (mg/L) of surfactin analogs was determined by the availability of the nitrogen source in the culture medium (surfactin A: from 0.04 (beef extract) to 2.14 (ammonium nitrate), surfactin B: from 1.59 (beef extract) to 52.52 (yeast extract), surfactin C: from 0.37 (beef extract) to 99.86 (yeast extract), surfactin D: from 0.87 (malt extract) to 15.05 (ammonium nitrate), surfactin E: from 2.14 (beef extract) to 147.24 (yeast extract), surfactin F: from 3.27 (beef extract) to 63.10 (yeast extract)).

The almost complete lack of surfactant activity of the strain growing in the beef extract medium was probably due to the lack of the isoform D and the low total concentration of surfactin analogs. Interestingly, the lack of isoform A among the analogs produced by the same strain in media with yeast extract and peptone did not translate into the significant deterioration of surface-active properties (Figure 3b).

### 2.3. Culture Conditions

Both the pH and the temperature of the culture significantly influenced the biosynthesis of surfactin by *B. subtilis natto* BS19 (Figure 5a–c and Figure 6a–c).

The highest (statistically significant) concentration of this compound was achieved at pH 7.0 and 37 °C. The increase in bacterial biomass was also the highest under these conditions. Strong reduction of surface tension (>50%) at different pH (6.0, 6.5, 7.0, 7.5 and 8.0) and different temperatures (30 °C, 36 °C, 37 °C, 38 °C and 40 °C) indicated that in each case the concentration of produced surfactin was probably higher than the critical micelle concentration. It can therefore be concluded that, although the composition of the medium is a key determinant of surfactin biosynthesis by *B. subtilis natto* BS19, the production of this compound can be significantly increased by appropriate selection of the culture conditions. Interestingly, pH of the medium significantly influenced the biosynthesis of individual surfactin isoforms. At pH 7.0, *Bacillus subtilis* synthesized mixtures of six analogs with a high proportion of isoforms C and E. The other pH values (6.0, 6.5, 7.5, 8.0) favored the production of analogs D and F (Figure 7).

In turn, the temperature of the culture did not have such a significant impact on the percentage of individual isoforms as pH did. At 36 °C and 37 °C, 4 or 5 surfactin analogs, respectively, were produced. At other temperatures tested, the production of 6 isoforms was observed (Figure 8).

### 2.4. Potato Peelings as an Alternative Carbon Source

Potato peel extract (PP) can successfully replace synthetic soluble starch as a component of the culture medium in surfactin biosynthesis. An increase in the biosynthesis of the mixture of surfactin isoforms and in the cell biomass of *B. subtilis natto* BS19 was observed, depending on the percentage of potato extract in the culture medium. The concentration of surfactin produced with the use of PP was similar to that obtained for soluble starch as a carbon source at the incubation temperature of 30 °C (Figure 6).

The surfactin concentration for PP was also higher than with the use of other analyzed carbon sources, except for the soluble starch in the variant with ammonium nitrate as the nitrogen source (Table 1, Figure 9). Additionally, the surface-active compounds formed as a result of microbiological synthesis lowered the surface tension of the PP medium by about 50% (Figure 9). The percentage of the five isoforms (Figure 10) in the media with different amounts of potato extract was similar.

Slight deviations were found in the medium containing 2% PP: the concentration of isoform B was lower and the concentration of isoform F higher than in the other variants. In the PP medium, the tested strain produced more isoform F and less isoform C compared to the medium containing synthetic starch. These differences are reflected in the surfactant activity of the synthesized mixture (ST reduction was >60% in starch medium as compared to >45% in PP medium (Figure 6 and Figure 9).

## 3. Discussion

The biosynthesis and properties of surfactants are determined by the genetic characteristics of the microorganism strain, the composition of the medium, and the growth conditions [28,37]. Therefore, the selection of nitrogen and carbon sources as well as the culture parameters influence the production of surfactin isoforms by the *Bacillus subtilis natto* BS19 strain. On the other hand, analysis of the trophic profile and metabolic activity of a given strain, enables the search for waste substrates that are a cheaper alternative to synthetic media.

In this study soluble starch, among the synthetic carbon sources we tested, turned out to be a carbohydrate not only conducive to the growth of *B. subtilis natto* BS19 (Figure 1), but also ensuring the highest efficiency of surfactin biosynthesis (Table 1). Other authors also demonstrated that this compound could be used as a substrate in the biosynthesis of surfactin and other lipopeptide biosurfactants [18,38,39,40]. *Bacillus* bacteria were shown to produce α-amylase, which randomly catalyzed the hydrolysis of α-1,4 glycosidic bonds inside the potato starch chain [41,42]. Hence, starch is a very good substrate for these bacteria. However, most researchers consider glucose and sucrose either as the optimal carbon source for the studied microorganisms [18,27], or as components that largely determine the effectiveness of surfactin biosynthesis [21]. Singh et al. [25] reported the maximum efficiency of biosurfactant production by the *Bacillus amyloliquefaciens* AR2 strain using sucrose as a carbon source (surface tension reduction 30–37 mM/m; critical micelle concentration (CMC) 80–110 mg/L; emulsification index (EI) (kerosene) 32–66%). The above data indicate that each strain exhibits its own ability to absorb individual carbohydrates. Interestingly, the *B. subtilis natto* BS19 strain produced more surfactin when grown on starch medium than in the presence of sucrose or glucose (Table 1) which is worth further analysis.

The presence of certain additional compounds in the media, such as growth stimulants (vitamins, amino acids) could be crucial for the development of a given strain [43]. The preferences for nitrogen source result from the need to assimilate a specific vitamin or amino acid, which may also translate into surfactin biosynthesis by *B. subtilis natto* BS19. Casein peptone, yeast extract, malt extract and beef extract differ in the qualitative and quantitative composition of amino acids, vitamins and carbohydrates. Casein peptone is an especially rich source of amino acids (tryptophan) and proteins [44]. Yeast extract contains carbohydrates, amino acids (especially glutamic acid), B vitamins (including biotin) and nucleotides [45]. Malt extract is also a rich source of carbohydrates (maltose, fructose, glucose and dextrins) and meat extract contains many B vitamins [46,47]. Both peptone and yeast extract turned out to be a better nitrogen source for *B. subtilis* BS19 than beef extract, malt extract or inorganic ammonium nitrate (Figure 3a–c). Similar results were reported by Cheng et al. [19], who supplemented *B. subtilis* cultures with yeast. They observed a 2% increase in bacterial biomass and found that the supplementation used was more effective in the production of biosurfactants than the use of fish or soy extract. Paraszkiewicz et al. [12] also used a medium supplemented with an extract of waste products (two different brewery wastewaters, beet molasses, apple peel extract), containing peptone and yeast extract. In turn, Abushady et al. [18] reported that inorganic nitrogen sources, such as ammonium nitrate and sodium nitrate, guaranteed a higher microbial production of surfactin (2.2 g/L and 1.9 g/L, respectively) than media containing yeast of peptone.

The culture medium pH and the temperature also influenced the synthesis of surfactin by *B. subtilis natto* BS19. The highest level of surfactin synthesis by the studied strain was observed at pH 7.0 and 37.0 °C (Figure 5c and Figure 6c). Chen et al. [20] showed that maintaining a constant pH enabled the biosynthesis of significant amounts of biosurfactants, including surfactin. Sen and Swaminathan [48] analyzed the effect of pH and temperature on the synthesis of surfactin by *B. subtilis* DSM 3256. The maximum production of surfactin (1.1 g/L) by the strain was obtained at 37.4 °C and pH 6.75. Abushady et al. [18] changed the pH of the medium from 5.0 to 9.0 in 0.5 steps. The highest concentration of surfactin (2.8 g/L) was found at pH 6.5, followed by 7 (2.4 g/L). A similar pH range (6.0–9.0) was analyzed by Abdel-Mawgoud et al. [28] in their study on the production of biosurfactants by *Bacillus subtilis* BS5. The highest increase in biomass they reached at pH 6.5–9.0, and the highest production of surfactin (2.25 g/L) at pH 6.8. In summary, the highest surfactin production is observed at pH of around 7.0 and 37 °C.

Surfactins are produced as a mixture of many analogs with a different structure, which translates into the biological and surface-active properties of this compound [10,40,49]. In this study, we confirmed that the composition of the medium and the culture conditions (especially pH) determined the number of isoforms produced and their percentage in the surfactin concentration (Figure 2, Figure 4, Figure 7, Figure 8 and Figure 10). However, there is still little work available on the biosynthesis of different surfactin variants in response to the modification of medium composition and culture conditions. Liu et al. [40] showed that the addition of various amino acids to the medium significantly affected the proportion of surfactin isoforms produced. When Arg, Gln or Val was added to the medium, more analogs with even β-hydroxy fatty acids were produced. In contrast, the addition of Cys, His, Ile, Leu, Met, Ser or Thr increased the proportion of surfactin variants with odd β-hydroxy fatty acids. Bartal et al. [37] presented a detailed study on the influence of the carbon source and Mn^2+^, Cu^2+^ and Ni^2+^ ions on the production of various surfactin analogs. They found that xylose and fructose had the greatest influence on the ratios of different analogs. Furthermore, Jajor et al. [10] showed that the culture conditions (aeration) of *B. subtilis* strains (KB1 and #309) influenced the proportions of the isoforms produced. A lower amount of oxygen decreased the proportion of the C15 isoforms in favor of the C12 analogs. Thus, biosynthesis of the desired surfactin analog depends on both the properties of the strain and the culture conditions.

In this study we tested the feasibility of using the selected waste product as a cheaper alternative to soluble starch. Commercialization of the production of surfactin analogs is currently difficult due to the high costs of their production. There are many cheaper alternatives to these products, such as sodium lauryl sulfate (SDS) which is a hundred times cheaper than the surfactin offered by Sigma Chemical Company. For this reason, efforts are made to reduce the cost of surfactin production. For example, attempts were made to reduce the expenditure on media components, which account for up to 50% of the total production costs of the compound [2,50]. We chose potato peelings because they are a waste product widely available in many countries. Importantly, potatoes are grown mainly in Europe and Asia [51]. Depending on the purpose of the treatment, they are usually peeled by the steam or abrasion methods. As a result, a large amount of peelings is created, the disposal of which is a serious problem for the industry. Annually, up to 140,000 tons of potato peelings are produced worldwide, most of which ends up in landfills [52]. Therefore, the use of this waste material not only reduces the cost of surfactin production, but also solves the problem of its management, which is desirable for environmental protection [53].

On the other hand, potato skins contain a wealth of various substances, including carbohydrates (mainly starch), vitamins and important elements [54]. We showed that the potato peel extract used promoted surfactin biosynthesis by *B. subtilis* BS 19. Additionally, the increase in the percentage of the extract in the medium translated into an increase in the production of surfactin (Figure 9c). Other studies demonstrated that most microorganisms of the genus *Bacillus* are hydrolytic to complex organic compounds [55]. Researches confirmed that starch-rich waste can be used as a substrate for the production of biosurfactants [32,56,57,58,59]. Ansari et al. [59] also analyzed the supplementation of the medium with nitrogen (sodium nitrate, urea). They reported that nitrogen supplementation stimulated the growth of microorganisms but had no effect on biosurfactant biosynthesis. The use of potato skins as an economic carbon source in the production of biosurfactants was also shown by Sharma et al. [53]. *Bacillus pumilus* DSVP18 strain used potato peelings as the only carbon source for the biosynthesis of biosurfactant (3.2 ± 0.32 g/L), which retained its surface-active properties in a wide range of temperature (20–120 °C), pH (2–12) and salinity (2–12%). Moreover, the microbial production of this biosurfactant resulted in a reduction of the surface tension of the medium from 72 to 28.7 nM/m. Waste potato peelings were also used in the microbial production of biosurfactants by Pande et al. [55]. Using this waste material, the *B. subtilis* DDU20161 strain synthesized 253.79 mg/L of surfactant after 40 h of culture.

In further studies with *B. subtilis natto* BS19, it is necessary to establish the structure as well as the surface-active and biological properties of the produced isoforms. In the next stage of the research, we intend to determine the composition of the medium and the growing conditions that enable the production of a surfactin isoform with the desired properties.

## 4. Materials and Methods

### 4.1. Microorganism

The native strain of *Bacillus subtilis natto* BS19, showing surfactin-producing ability, was isolated from a food product named natto by a multi-stage screening process as described by Koim-Puchowska et al. [60]. Bacterial isolates were identified by sequence analysis of the 16S rRNA gene. The research material was deposited into a cryobank (Grasso Biotech, Starogard Gdańsk, Poland) and stored at −20 °C until the analysis. Before the experiments, pure cultures of *B. subtilis natto* BS19 were transferred from the cryobank, plated on an agar plate, and then grown on agar slants (T = 37 °C; t = 72 h). The cultures on agar slants were stored at +4 °C.

### 4.2. The Media and Culture Conditions

Before the mineral medium was inoculated, the *B. subtilis natto* BS19 strain was grown in nutrient broth (5 g/L tryptone peptone, 2.5 g/L yeast extract, 1 g/L glucose, pH 7.2–7.4) for 24 h at rpm = 70. Mineral medium [13], containing 40 g/L glucose, 4 g/L NH_4_NO_3_, 4.08 g/L KH_2_PO_4_, 7.12 g/L Na_2_HPO_4_ × 2H_2_O, 0.2 g/L MgSO_4_ × 7H_2_O, 0.0008 g/L CaCl_2_, 0.0011 g/L FeSO_4_ × 7H_2_O, 0.0012 g/L ethylenediaminetetraacetic acid (EDTA) and pH 7.0, was inoculated with 2.5 mg of bacteria. The mineral medium was modified using various synthetic sources of carbon and nitrogen, which resulted in numerous variants of the medium. In the first stage of the research, 14 carbon sources, differing in structure and chemical properties, were examined in order to learn about the trophic profile of the strain under study and the usefulness of individual compounds in the microbiological synthesis of surfactin. As synthetic carbon sources we used monosaccharides (arabinose, xylose, fructose, glucose, mannose), disaccharides (cellobiose, galactose, lactose, maltose, sucrose, trehalose) and soluble starch (a polysaccharide). Two sugar alcohols were also tested: glycerin and sorbitol. In the first stage of the research, inorganic ammonium nitrate was used as the nitrogen source in accordance with the original composition of the culture medium [13]. Hence, in the second stage of the study, four organic nitrogen sources were used as an alternative to ammonium nitrate: yeast extract, beef extract, malt extract and casein peptone. All these nitrogen sources have been used so far in surfactin biosynthesis [21]. Having selected the optimal carbon and nitrogen source for *B. subtilis natto* BS19, in the next two stages of the research, we focused on determining the effect of culture conditions (culture pH and temperature) on surfactin biosynthesis. Cultures with shaking (70 rpm) were carried out in triplicate in flasks (v = 250 mL) for 120 h at four different pH values, 6.0, 6.5, 7.5 and 8.0, in addition to the previously tested variant at pH 7.0 and 37 °C. After verification of this parameter, the cultures were additionally carried out at four other temperatures: 30 °C, 36 °C, 38 °C and 40 °C. In the last stage of the study, the use of an aqueous extract of potato peelings (PP) as an alternative to a synthetic carbon source in surfactin biosynthesis was assessed. To this end, the potato peelings were rinsed three times with distilled water and dried at 55 °C for 96 h, and then ground in an analytical mill. To obtain the potato extract, an aqueous suspension 10% (*w*/*v*) of the dried material was autoclaved (T = 121 °C, t = 15 min), filtered through gauze and then through paper filter under reduced pressure. Potato peel water extract was added to the culture media (2%, 4% or 6%) instead of the synthetic carbon source [32,61].

### 4.3. Determination of Bacillus Subtilis Natto BS19 Biomass Concentration

In order to measure the optical density of the cell suspension of *B. subtilis natto* BS19, 5 mL of the material were taken from each flask (in duplicate) and then centrifuged (MPW-260R laboratory centrifuge, MPW-Med. Instruments, Warszawa, Poland). After decanting the supernatant, the material was suspended in sterile 0.9% NaCl, vortexed (t = 30 s) and centrifuged again; this was repeated twice to completely remove the culture medium from the sample. The optical density (OD600) of the cell suspension in sterile 0.9% NaCl was measured using a UV-vis spectrophotometer (Pharo 300, Merck). The cell biomass concentration (mg/mL) in inoculum and post-culture medium was determined using a curve showing the relationship between cell biomass dry weight and optical density (OD). The curve was prepared with a pure line culture of *Bacillus subtilis* no. ŁO820 from the Collection of Industrial Strains, Łódź University of Technology, Łódź, Poland. The biomass was dried to a constant mass using a weighting dryer (RADWAG WPS-30S, Radwag, Radom, Poland) at 105 °C and 20 s sampling time [60].

### 4.4. Reduction of the Surface Tension of the Culture Medium

The reduction of the surface tension of the medium during the culture was determined as the difference in the surface tension measured before inoculation and after the culture was completed. The surface tension was measured in triplicate by the du Noüy ring method using a tensiometer (PI-MT1M model, Donserv, Warszawa, Poland). Measurements performed after the culture was completed were preceded by the removal of the cell biomass from the culture medium by centrifugation (2400× *g*, t = 15 min, T = 4 °C [62].

### 4.5. Surfactin Extraction and Determination of Its Concentration

Affinity chromatography with a SPE (solid-phase extraction) system was used to extract surfactin isoforms. In the process we used Agilent Technologies Bond Elut C18 columns (Santa Clara, CA, USA), dedicated to the isolation of hydrophobic compounds. A detailed procedure of surfactin extraction was previously described in Koim-Puchowska et al. [60]. The surfactin concentration was determined by high performance liquid chromatography (HPLC) using an Agilent Technologies model 1220 device equipped with a diode detector. The chromatographic separation was carried out under the following conditions: Poroshell 120 EC-C18 column (4.6 × 150 mm), mobile phase 80:20 (acetonitrile: 3.8 mM trifluoroacetic acid), 40 °C, detection at 205 nm. The concentration of surfactin was calculated as the sum of the peaks of all 6 isoforms using the external standard (ESTD) method. The concentration of surfactin isoforms was determined by HPLC using surfactin provided by Sigma Aldrich (St. Louis, MO, USA) (Surfactin from *Bacillus subtilis*, ≥98.0%—S3523).

### 4.6. Measurement of the Hydrolytic Activity of Microorganisms

Tests evaluating the hydrolytic activity of *B. subtilis natto* BS19 against various types of substrates were carried out on agar media (tryptone peptone 5 g/L, yeast extract 2.5 g/L, glucose 1 g/L, 15 g/L agar, pH 7.2–7.4, containing 1% soluble starch, 1% cooking oil, or 1% skim milk), and on Vogels agar with 1% low viscosity sodium carboxymethylcellulose (CMCNa) (1% CMCNa, 5 g/L Na_3_C_6_H_5_O_7_, 5 g/L KH_2_PO_4_, 2 g/L NH_4_NO_3_, 4 g/L (NH_4_)_2_SO_4_, 0.2 g/L MgSO_4_, 1 g/L peptone, 2 g/L yeast extract, 15 g/L agar, pH 5.5). Solid media were inoculated with *B. subtilis natto* BS19 using a sterile loop. Agar plates were incubated for 48 h at 30 °C. The surfaces of the agar plates containing 1% soluble starch were then flooded with Lugol’s iodine for 1–2 min. The surfaces of the agar plate supplemented with 1% CMCNa were flooded with 0.1% aqueous solution of Congo red for 30 min; after decanting the reagent, the surface of the plate was flooded with 1 M NaCl aqueous solution for 10 min. After performing the above-mentioned tests, all substrates were examined for the presence of clear hemolysis zones [10,63].

### 4.7. Statistical Analysis

The experiments were performed in triplicate (*n* = 3). Results are presented as mean ± standard deviation (SD). The analysis of variance (ANOVA) and Tukey’s test for equal groups were used to verify the influence of the substrate composition and culture conditions on: (i) *B. subtilis natto* BS19 biomass concentration, (ii) reduction of the surface tension of the medium during the culture and (iii) the final concentration of the synthesized surfactin isoforms. Statistica v. 13.3 (TIBCO Software Inc. Palo Alto, CA, USA) and Excel software were used to analyze the obtained data.

## 5. Conclusions

The presented results confirm that the medium composition largely determines the growth of *B. subtilis natto* BS19 cell biomass and the biosynthesis of surfactin, which translates into differences in the reduction of the surface tension of the medium. The highest efficiency of surfactin production was found with the use of soluble starch as a carbon source (368.36 ± 94.23 mg/L), although most authors point to glucose or sucrose. On the other hand, supplementation of the medium with an organic nitrogen source (yeast extract, peptone) instead of ammonium nitrate used in Cooper’s medium resulted in about 5 times higher concentration of cell biomass; surfactin biosynthesis in media with peptone and yeast extract increased two- and three-fold, respectively. Factors such as temperature and pH of the culture also influenced the cell biomass, surface tension and surfactin concentration. The best results in the production of this compound were observed at pH 7.0 and 37 °C. Potato peel extract proved to be a cheaper alternative to soluble starch. A significant influence of this substrate on the production of surfactin was confirmed by the increase in the average cell biomass of *B. subtilis natto* BS19; it was also observed that the higher the percentage of extract in the medium, the higher the surfactin biosynthesis (mean biomass concentration (mg/mL)/mean surfactin concentration (mg/L): 2%/1.28/30.93/; 4% /2.01/45.96/; 6%/2.22/58.2/. Regardless of the amount of potato peel extract in the medium (2%, 4%, 6%), a reduction in the surface tension of the medium was observed (up to 50%). This suggests that in all experimental variants this compound was produced at the CMC level. *Bacillus subtilis natto* BS19 is a strain capable of biosynthesis of various surfactin isoforms. The percentage of each isoform is largely determined by the composition of the medium and especially pH of the culture. Further studies are necessary to investigate the structure of individual isoforms that determine the biological or surface-active properties of the compound. Understanding the relationship between the medium composition and the synthesis of the surfactin analog with the desired properties will allow the production of this compound for a specific purpose; in the future it will translate into lower production costs.

## Figures and Tables

**Figure 1 molecules-26-02985-f001:**
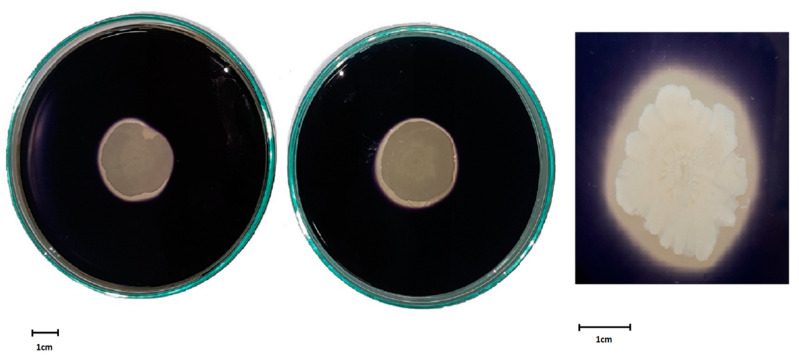
Clear zone around a single colony of *B. subtilis natto* BS19 on agar medium supplemented with 1% soluble starch after staining with Lugol’s iodine.

**Figure 2 molecules-26-02985-f002:**
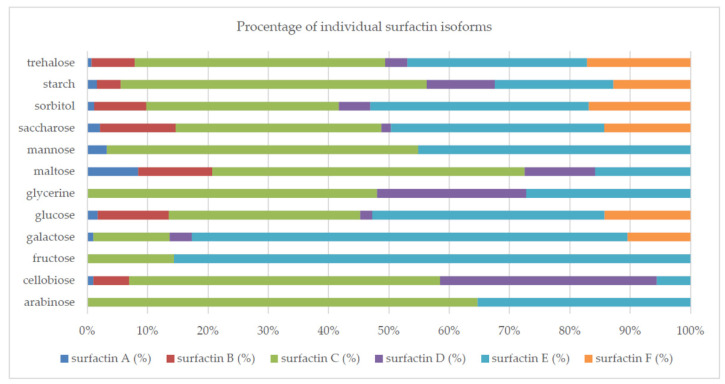
Percentage of individual surfactin isoforms for different carbon sources.

**Figure 3 molecules-26-02985-f003:**
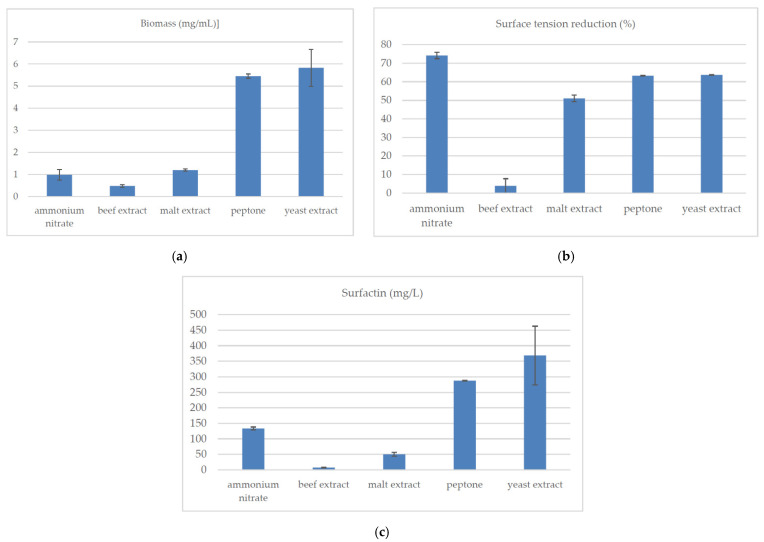
Influence of nitrogen source on (**a**) the concentration of *B. subtilis natto* BS19 biomass (mg/mL), (**b**) reduction of surface tension (%) of the medium during the culture and (**c**) concentration of surfactin (mg/mL). Data presented as mean ± SD for each nitrogen source (*n* = 3). The mean values given in lines with different letter indices are significantly different (α ≤ 0.05). Composition of culture medium: soluble starch (40 g/L), NH_4_NO_3_/beef extract/malt extract/peptone/yeast extract (4 g/L), KH_2_PO_4_ (4.08 g/L), Na_2_HPO_4_ × 2H_2_O (7.12 g/L), MgSO_4_ × 7H_2_O (0.2 g/L), CaCl_2_ (0.0008 g/L), FeSO_4_ × 7H_2_O (0.0011 g/L), ethylenediaminetetraacetic acid (0.0012 g/L).

**Figure 4 molecules-26-02985-f004:**
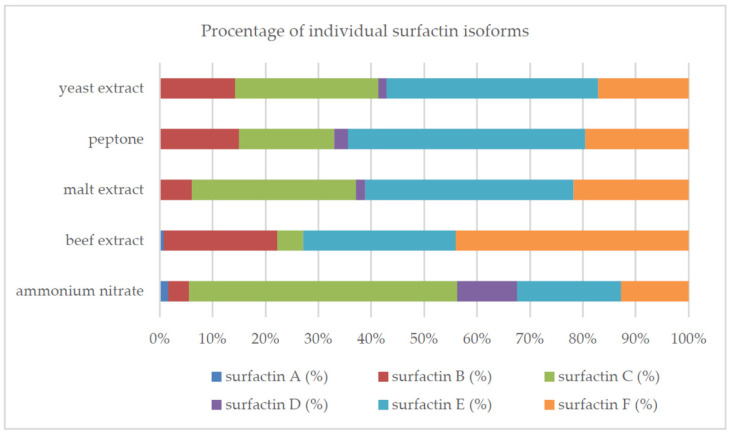
Percentage of individual surfactin isoforms for different nitrogen sources.

**Figure 5 molecules-26-02985-f005:**
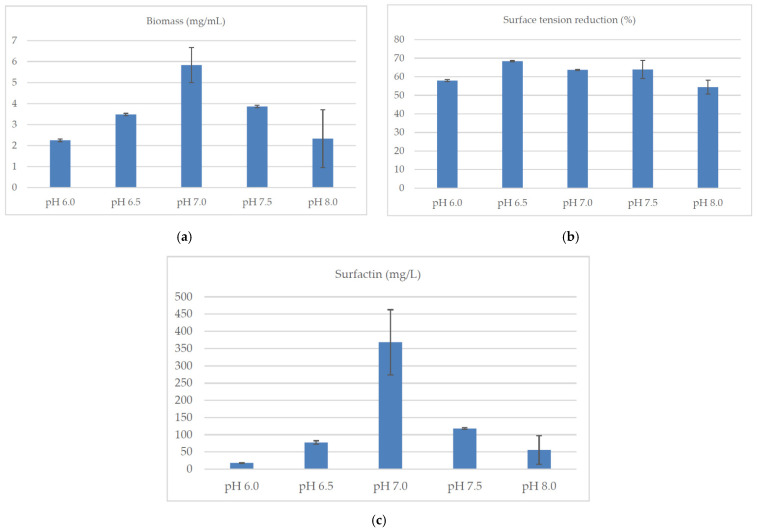
Influence of pH of the culture medium on (**a**) the concentration of *B. subtilis natto* BS19 biomass (mg/mL), (**b**) reduction of surface tension (%) of the medium during the culture and (**c**) concentration of surfactin (mg/mL). Data presented as mean ± SD (*n* = 3). The mean values given in lines with different letter indices are significantly different (α ≤ 0.05). Composition of culture medium: soluble starch (40 g/L), yeast extract (4 g/L), KH_2_PO_4_ (4.08 g/L), Na_2_HPO_4_ × 2H_2_O (7.12 g/L), MgSO_4_ × 7H_2_O (0.2 g/L), CaCl_2_ (0.0008 g/L), FeSO_4_ × 7H_2_O (0.0011 g/L), ethylenediaminetetraacetic acid (0.0012 g/L).

**Figure 6 molecules-26-02985-f006:**
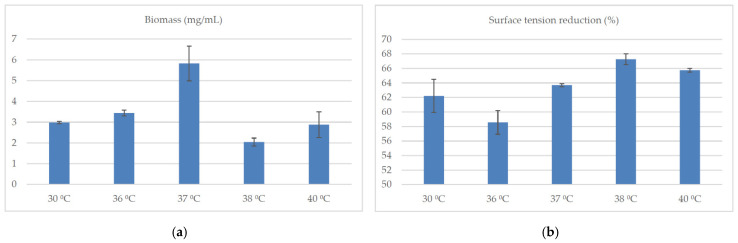
Influence of the temperature of the culture medium on (**a**) the concentration of *B. subtilis natto* BS19 biomass (mg/mL), (**b**) reduction of surface tension (%) of the medium during the culture and (**c**) concentration of surfactin (mg/mL). Data presented as mean ± SD (*n* = 3). The mean values given in lines with different letter indices are significantly different (α ≤ 0.05). Composition of culture medium: soluble starch (40 g/L), yeast extract (4 g/L), KH_2_PO_4_ (4.08 g/L), Na_2_HPO_4_ × 2H_2_O (7.12 g/L), MgSO_4_ × 7H_2_O (0.2 g/L), CaCl_2_ (0.0008 g/L), FeSO_4_ × 7H_2_O (0.0011 g/L), EDTA (0.0012 g/L).

**Figure 7 molecules-26-02985-f007:**
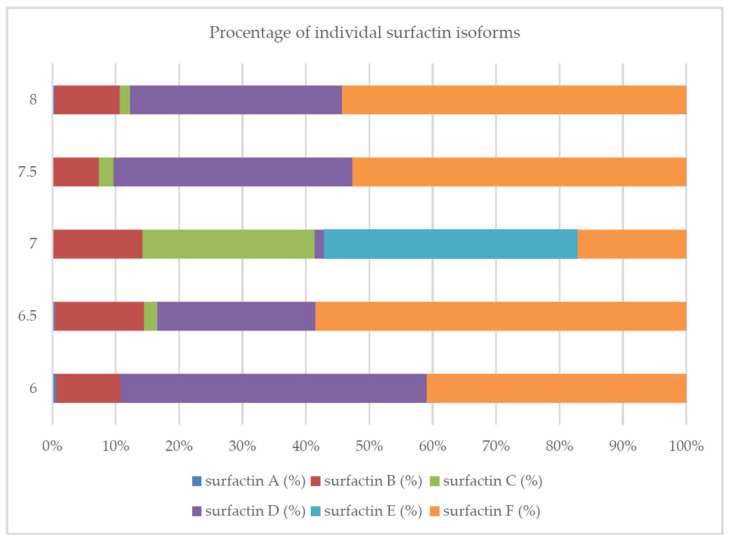
Percentage of individual surfactin isoforms for different pH of medium.

**Figure 8 molecules-26-02985-f008:**
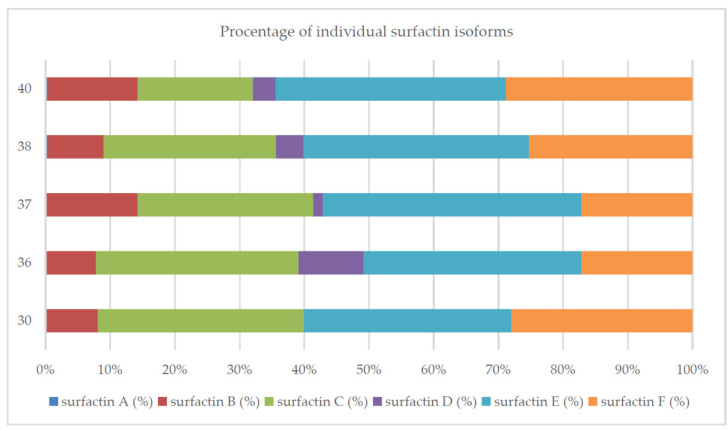
Percentage of individual surfactin isoforms for different temperatures of culture.

**Figure 9 molecules-26-02985-f009:**
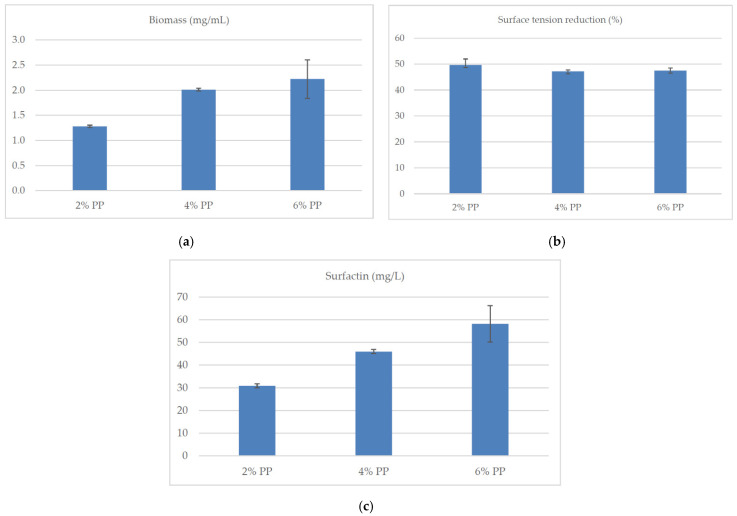
Influence of the alternative carbon source (potato peel extract) on (**a**) the concentration of *B. subtilis natto* BS19 biomass (mg/mL), (**b**) reduction of surface tension (%) of the medium during the culture and (**c**) concentration of surfactin (mg/mL). Data presented as mean ± SD (*n* = 3). The mean values given in lines with different letter index are significantly different (α ≤ 0.05). Composition of culture medium: potato peel extract (PP) (2%/4%/6%), yeast extract (4 g/L), KH_2_PO_4_ (4.08 g/L), Na_2_HPO_4_ × 2H_2_O (7.12 g/L), MgSO_4_ × 7H_2_O (0.2 g/L), CaCl_2_ (0.0008 g/L), FeSO_4_ × 7H_2_O (0.0011 g/L), ethylenediaminetetraacetic acid (0.0012 g/L).

**Figure 10 molecules-26-02985-f010:**
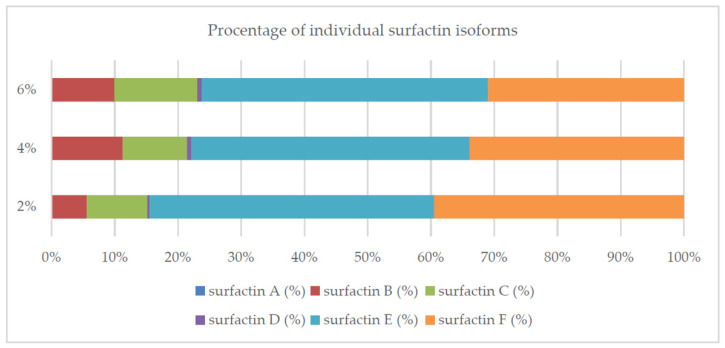
Percentage of individual surfactin isoforms for different amounts of potato extract added to the culture medium.

**Table 1 molecules-26-02985-t001:** Influence of carbon source on the concentration of *B. subtilis natto* BS19 biomass (mg/mL), reduction of surface tension (%) of the medium during the culture and concentration of surfactin (mg/mL).

Source of Carbon	Biomass (mg/mL)	Surface Tension Reduction (%)	Surfactin (mg/L)
	Mean ± SD	Mean ± SD	Mean ± SD
arabinose	0.42_abdfjlmn_ ± 0.12	2.23_e_ ± 2.23	0.29_d_ ± 0.01
cellobiose	1.19_abcdehij_ ± 0.02	25.06_dg_ ± 1.29	4.09_ad_ ± 0.34
fructose	0.68_abcde_ ± 0.03	17.30_c_ ± 0.78	0.77_d_ ± 0.21
galactose	0.58_abcdfijl_ ± 0.04	29.85_afgh_ ± 3.64	8.80_acd_ ± 1.34
glucose	0.84_abcde_ ± 0.08	44.03_b_ ± 1.64	3.68_ad_ ± 1.81
glycerine	1.7_ace_ ± 0.90	31.70_ad_ ± 1.43	1.17_d_ ± 0.92
lactose	0.1_bdf_ ± 0.00	4.27_e_ ± 4.27	0.00_d_ ± 0.00
maltose	1.21_abcde_ ± 0.10	36.37_a_ ± 1.54	1.00_d_ ± 0.25
mannose	1.2_abcdehi_ ± 0.06	20.22_cf_ ± 4.04	0.87_d_ ± 0.31
saccharose	1.47_abceh_ ± 0.07	30.09_ad_ ± 0.22	1.88_ad_ ± 1.41
sorbitol	1.03_abcde_ ± 0.17	35.37_a_ ± 0.11	12.71_ac_ ± 9.04
starch	0.98_abcdehijkln_ ± 0.24	74.15_b_ ± 1.70	133.38_b_ ± 4.83
trehalose	1.56_abcehijk_ ± 0.14	33.20_ah_ ± 2.73	17.98_ac_ ± 4.71
xylose	0.05_dflm_ ± 0.01	0.00_e_ ± 0.00	0.00_d_ ± 0.00

Data presented as mean ± SD (*n* = 3). The mean values given in lines with different letter indices are significantly different (α ≤ 0.05). Composition of culture medium: arabinose/xylose/fructose/glucose/mannose/cellobiose/galactose/lactose/maltose/sucrose/trehalose/soluble starch (40 g/L), NH_4_NO_3_ (4 g/L), KH_2_PO_4_ (4.08 g/L), Na_2_HPO_4_ × 2H_2_O (7.12 g/L), MgSO_4_ × 7H_2_O (0.2 g/L), CaCl_2_ (0.0008 g/L), FeSO_4_ × 7H_2_O (0.0011 g/L), ethylenediaminetetraacetic acid (0.0012 g/L).

## Data Availability

The data presented in this study are available on request from the corresponding author.

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
