# Peer review of "Influence of the Medium Composition and the Culture Conditions on Surfactin Biosynthesis by a Native Bacillus subtilis natto BS19 Strain"

_molecules, 2021, doi:10.3390/molecules26102985_

Round 1

Reviewer 1 Report

The paper is interesting and the experiments adequately planned. Minor revision for the English language are requested. The Introduction is exhaustive and well written. It provides an adequately clear picture of the topic and on the currently available knowledge. Materials and methods have to be improved in some points to make more clear all the experimentation. Some suggestions:

Line 11. producer strain

Line 98, At various temperature and pH conditions

Line 105. The name of the strain in italics

Lines 124-125. Please, explain better, is not the glucose to produce surfactin but the strain grown in the presence of it

Line 135. To test the effect of nitrogen sources, which conditions you have assessed for the other parameters?

Line 166. The name of the strain in italics. Please check throughout the manuscript

Materials and methods.

Here it is not clear if you have used a single strain or more strains.

Lines 354-364. Probably this section is more suitable for the discussions

Line 379. B. subtilis in italics

Line 388. the name of the strin in italics

Two points are not enough clear and specified: how did you distinguish the different surfactin isoforms; the culture approach: after carbon sources screening, which carbon source have you maintaned to test nitrogen sources? the best one resulted from the previous screening? The same question for T and pH tests.

The Discussion section could be improved, by focusing more on the use of alternative and cheaper sources (potato peel extract) that is the real new aspect of the paper.  A more concrete and critical discussion on whether its use can actually reduce production costs would be useful.

I suggest to accept the manuscript after minor revisions.      

Reviewer 2 Report

This manuscript does what the well written abstract promises and deserves it to be published, eventually. To attain publication quality I have the following suggestions:

  • The authors should describe the external standards used to identify and quantify the individual surfactins.
  • The authors should shorten the discussion and start with and focus on the main results; the abstract is clear; the discussion should be structured accordingly.
  • The results section contains several Tables where it is not immediately clear what/where the optimum is. It would be better to present these data as Figures. In many cases bar diagrams seem the best option; in other cases line diagrams are the better option.

Minor comments

L20: less decimals.

L33 and L35: i.e. should read viz.

L35: subtilis (typo)

L38-39: this sentence implies that surfactins are a group/class of compounds and suggests that the example given in line 1 is for one specific surfactin. The introduction should be rewritten to reflect  this.  Throughout the present ms, especially in the introduction, the authors erroneously write about “this compound” where it is actually these compounds. In line 96, by stressing isoforms,  it is clear that they know better than that.

L96: the aim is to quantify isoforms; in the subsequent paragraph quantification of isoforms is not mentioned?

L105: organism name in italics (typo).

Table 1: specify the number of replicate cultures per carbon source.

Table 1: less decimals for the SDs.

Figure 2 and 3: specify the composition of the various isoforms or refer to the place in the ms where this will be done.

Table 2: this data could be better presented as a column diagram.

Table 2: indicate in a footnote what was the carbon source with ammonium nitrate. I take it that the other substrates were used as is, without an additional carbon source?

Line 144-146: too many decimals.

Table 3 and Table 4: give the growth substrate.

L175-186: this sections shows several cases where the name of the organism was not italicised; make sure that this is corrected throughout this paragraph and the ms as a whole.

L223: growth rather than growing.

Round 2

Reviewer 2 Report

I am pleased with the revisions and am happy to see this published.